# The Synergic Effect of AT(N) Profiles and Depression on the Risk of Conversion to Dementia in Patients with Mild Cognitive Impairment

**DOI:** 10.3390/ijms24021371

**Published:** 2023-01-10

**Authors:** Marta Marquié, Fernando García-Gutiérrez, Adelina Orellana, Laura Montrreal, Itziar de Rojas, Pablo García-González, Raquel Puerta, Clàudia Olivé, Amanda Cano, Isabel Hernández, Maitée Rosende-Roca, Liliana Vargas, Juan Pablo Tartari, Ester Esteban-De Antonio, Urszula Bojaryn, Mario Ricciardi, Diana M. Ariton, Vanesa Pytel, Montserrat Alegret, Gemma Ortega, Ana Espinosa, Alba Pérez-Cordón, Ángela Sanabria, Nathalia Muñoz, Núria Lleonart, Núria Aguilera, Ainhoa García-Sánchez, Emilio Alarcón-Martín, Lluís Tárraga, Agustín Ruiz, Mercè Boada, Sergi Valero

**Affiliations:** 1Ace Alzheimer Center Barcelona, Universitat Internacional de Catalunya (UIC), 08028 Barcelona, Spain; 2Networking Research Center on Neurodegenerative Diseases (CIBERNED), Instituto de Salud Carlos III, 28029 Madrid, Spain

**Keywords:** mild cognitive impairment (MCI), dementia, CSF, NPSs, interaction, synergic

## Abstract

Few studies have addressed the impact of the association between Alzheimer’s disease (AD) biomarkers and NPSs in the conversion to dementia in patients with mild cognitive impairment (MCI), and no studies have been conducted on the interaction effect of these two risk factors. AT(N) profiles were created using AD-core biomarkers quantified in cerebrospinal fluid (CSF) (normal, brain amyloidosis, suspected non-Alzheimer pathology (SNAP) and prodromal AD). NPSs were assessed using the Neuropsychiatric Inventory Questionnaire (NPI-Q). A total of 500 individuals with MCI were followed-up yearly in a memory unit. Cox regression analysis was used to determine risk of conversion, considering additive and multiplicative interactions between AT(N) profile and NPSs on the conversion to dementia. A total of 224 participants (44.8%) converted to dementia during the 2-year follow-up study. Pathologic AT(N) groups (brain amyloidosis, prodromal AD and SNAP) and the presence of depression and apathy were associated with a higher risk of conversion to dementia. The additive combination of the AT(N) profile with depression exacerbates the risk of conversion to dementia. A synergic effect of prodromal AD profile with depressive symptoms is evidenced, identifying the most exposed individuals to conversion among MCI patients.

## 1. Introduction

Neuropsychiatric symptoms (NPSs), defined as behavioral or psychological symptoms, include impairment in mood, anxiety, drive, perception, sleep as well agitation and aggression [1]. NPSs have been frequently associated with poorer outcomes in cognition, functional state, quality of life and rate of progression to severe dementia [2,3,4]. In fact, NPSs have also been considered an early manifestation in the preclinical and prodromal mild cognitive impairment (MCI) stages of Alzheimer’s disease (AD), predicting a higher risk of conversion to dementia [5].

In recent decades, several in vivo biomarkers able to quantify the main AD-related changes present in the brain (Aβ-amyloid (Aβ) plaques, neurofibrillary tangles composed of hyperphosphorylated tau, neurodegeneration and vascular changes) have been developed, including fluids (cerebrospinal fluid (CSF), plasma), imaging (magnetic resonance, positron emission tomography (PET)) and genomic tools.

In 2018, the National Institute of Aging–Alzheimer’s Association (NIA–AA) research criteria for AD were reviewed to include AD-related biomarkers [6]. The resulting AT(N) framework, intended for research use, established a biological definition of AD independently of the clinical syndrome presented by the patient. It categorized AD-related brain changes that could be detected in vivo using different imaging and fluid biomarkers into three groups (Aβ plaques (A), pathologic tau (T) and neurodegeneration (N)). Thus, individuals that are not demented (cognitively unimpaired or with MCI) could be classified as presenting underlying AD pathology if they exhibited abnormal A and T biomarkers.

To date, few studies have focused on the association between AD-core biomarkers and NPSs in patients with MCI, and none on the interaction effect of the AT(N) profile and NPSs on the risk of conversion to dementia.

The main aim of the present study is to explore the predictive value of the combination of the AT(N) profile (quantified using CSF AD-core biomarkers) and NPSs (assessed by the Neuropsychiatric Inventory Questionnaire (NPI-Q) [7] using survival analysis to determine the conversion ratio to dementia in a sample of MCI patients followed-up in a memory clinic. Additionally, we sought to examine the convergence effect of the AT(N) profile and NPSs to determine their synergic and/or multiplicative interactions on the estimation of conversion to dementia.

## 2. Results

The demographic and clinical characteristics of the study sample are shown in Table 1. The mean age of the participants was 73 years, 55% were females, with an average of formal education of 8 years.

Regarding the classification of participants into the four AT(N) groups, prodromal AD was the most common (42%), followed by normal (26%), SNAP (18%) and lastly brain amyloidosis (14%).

The most frequent NPS present in the study sample was depression (50%) followed by anxiety (44%), while nighttime behaviors exhibited the lowest prevalence (20%).

It was found that 80% of the participants had an amnestic MCI subtype and 42% a probable MCI status. Lastly, 36% were *APOE* ε4 carriers.

Several demographic and clinical variables were significantly associated with AT(N) profiles. The prodromal AD group presented an older age (*p* < 0.001), a highest percentage of amnestic (*p* = 0.001) and probable (*p* < 0.001) MCI subtypes, the presence of the *APOE* ε4 allele (*p* < 0.001) and lower MMSE scores (*p* = 0.026), as expected (Table 2).

Nighttime behaviors was the only NPS that significantly differed among the four AT(N) profiles (*p* = 0.023) (Table 3), exhibiting the lowest prevalence in the prodromal AD group.

When exploring the characteristics of the different NPS groups (Table 4), we found that depression and anxiety were more common in females (*p* < 0.001), while irritability was more common in males (*p* < 0.001). The *APOE* ε4 status did not show a significant association with the presence of any NPS. Nighttime behaviors was the only NPS significantly associated with the AT(N) profiles, that is, the prodromal AD group was more prevalent among patients without this NPS (*p* < 0.05) (Table 3).

During the follow-up period, 224 MCI participants (44.8% of the whole sample) converted to dementia. Compared to MCI non-converters, MCI converters were older (*p* < 0.001), had more frequently amnestic (*p* < 0.001) and probable (*p* < 0.001) profiles, lower MMSE scores (*p* < 0.001) and a higher frequency of the *APOE* ε4 allele (*p* = 0.01). Apathy was the only NPS that differed between both groups, exhibiting a higher frequency in MCI converters (*p* = 0.029). Lastly, with respect to the AT(N) groups distribution, close to 60% of MCI converters had a prodromal AD profile, compared to 26% of the MCI non-converters (*p* < 0.001) (Table 4).

A Cox regression analysis of risk conversion, with AT(N) profiles and the NPSs as main factors, is reported in Table 5. Regarding the AT(N) profiles, the prodromal AD, brain amyloidosis and SNAP groups emerged as risk factors of conversion to dementia compared to the normal group (HR = 4.34, 2.92 and 1.78, respectively) (see Figure 1). When considering NPSs, a higher risk of conversion to dementia was observed only in patients with the presence of depression and apathy symptoms (HR = 1.46 and 1.4, respectively).

An additive interaction between AT(N) profiles and NPSs was detected for depression (*p* = 0.037, Table 6). The increased risk of conversion to dementia among patients with depression, compared to those without, is higher in the prodromal AD AT(N) group than in the normal AT(N) group.

Secondly, a multiplicative interaction effect was observed between AT(N) profiles and the NPS nighttime behaviors, but in an opposite sense than the former (Table 6). In patients in the normal AT(N) group, in comparison with those in the prodromal AD AT(N) group, the presence of nighttime behaviors was a predictor of a higher risk of conversion to dementia (*p* = 0.034).

Having in mind that comorbidities are frequent among NPSs in patients with cognitive impairment (meaning that NPSs do not usually occur isolated but accompanied by other NPSs), the same additive interaction for AT(N)*Depression and multiplicative interaction for AT(N)*Nighttime Behaviors detected previously were explored again, now adding to the regression model the other four NPSs as covariates. Under this adjustment, the multiplicative interaction for ATN*Nighttime Behavior lost its significance, but the additive interaction for AT(N)*Depression remained statistically associated with conversion to dementia (*p* = 0.044). Figure 2 shows the differential impact of the NPS depression on the cumulative hazard function depending on the AT(N) profiles.

## 3. Discussion

In this study we analyzed the impact of NPSs, assessed using the NPI-Q [7], and the AT(N) profiles, quantified in CSF, on conversion to dementia in a cohort of 500 MCI patients who were longitudinally followed-up in a memory clinic for 2 years. Our results indicate that all pathologic AT(N) groups (brain amyloidosis, prodromal AD and SNAP) but also two of the NPSs (depression and apathy) are high risk factors of conversion to dementia. Moreover, an additive interaction effect between the AT(N) profile and depression on conversion to dementia was identified. To the best of our knowledge, this is the first study exploring the modulating effect of AT(N) profiles and neuropsychiatric status of MCI patients in predicting conversion to dementia.

Confirming the results of a recent publication from our group exploring a larger sample of MCI participants [8], the AT(N) profiles emerged here as significant discriminant factors in the prediction of conversion to dementia. Most of the MCI patients who converted to dementia (60%) belonged to the prodromal AD AT(N) profile and this group was by far the strongest predictive condition of conversion to dementia in our study (HR = 4.34 compared to the normal AT(N) group). The other two pathological AT(N) profiles (brain amyloidosis and SNAP) presented intermediate ratios of conversion to dementia (SNAP: HR = 1.78, rain amyloidosis: HR = 3, both compared to the normal AT(N) group). In contrast, MCI patients within the normal AT(N) group showed a 12% conversion to dementia. It is important to highlight the strength of the results of this study, especially having in mind the short longitudinal follow-up time in our study (2 years).

Among the NPSs, only depression and apathy, assessed by the NPI-Q [7] at baseline, predicted conversion to dementia. We believe that the effects of these two NPSs are very relevant and confer a specific and real prediction risk, considering all the relevant clinical and biological factors accounted for in the statistical model (CSF AD-core biomarkers, *APOE* ε4 status and amnestic and probable MCI subtypes). Moreover, and reinforcing the specificity of these results, depression and apathy are frequently presented in a comorbid way; that is, these two symptoms are usually co-occurring in the same individual [9]. In the present study, however, the effect of depression and apathy were mutually adjusted, as were the rest of the NPSs. In this sense, a previous study from our group [3] showed somehow different results, but without considering the potential comorbid presentation of NPSs. In this work, using data from over 2100 MCI patients, NPSs were assessed by NPI-Q at baseline and classified into four profiles (irritability, apathy, anxiety/depression and asymptomatic) using latent class analysis. Irritability and apathy were both predictive of conversion to dementia, while anxiety/depression showed no risk compared to the asymptomatic class.

Neuropathological series show that “mixed” dementia (the combination of cerebrovascular disease and neuropathological changes of AD mostly) is the most frequent cause of cognitive impairment in the elderly [10]. Additionally, MCI cases with “mixed” pathology are at more risk of conversion to dementia than those with “pure AD” changes [11]. Interestingly, depression and apathy have been suggested to present an underlying cerebrovascular etiology [12,13]. Thus, cerebrovascular dysfunction could be a good explanation for the connection between NPSs and AD [14]. In fact, previous studies have identified depression-spectrum symptoms, including apathy, as risk factors for increased vulnerability to AD pathophysiologic changes and clinical decline [15], as well as predictors of greater atrophy in AD-related brain regions in MCI patients [16]. Other studies have suggested that depression could not only precede but also accelerate cognitive decline in AD patients [17,18]. Importantly for our results, a prodromal AD AT(N) profile in CSF does not exclude the presence of underlying “mixed” dementia in a given case, as the AT(N) classification does not take into consideration cerebrovascular changes [6].

Our results show that the most important factor for conversion to dementia in MCI was the modulating effect of combining the AT(N) profiles with NPSs. Depression was the only NPS that emerged as significant under this approach, and the obtained modulating effect seemed to be optimally explained assuming an additive process. On close examination, this synergistic effect meant that the risk of conversion to dementia among MCI patients with depressive symptoms (compared to patients without) was increased in the prodromal AT(N) group, compared to the normal AT(N) group (thus, the risk of conversion to dementia was higher in patients in the prodromal AD group with depressive symptoms than expected by the sum of the two main exposure effects, namely AT(N) profile and depressive symptoms separately). Additionally, a multiplicative interaction was also observed for the NPS nighttime behavior when this was the only NPS considered in the analysis. In this case, however, the modulating effect had a different interpretation. We detected a significantly higher risk of conversion to dementia in patients with nighttime behavior and a normal AT(N) profile, which disappeared in those within the prodromal AT(N) group. However, the modulating effect of the AT(N) profile on this symptom became irrelevant when considering the comorbid effect of the rest of the NPSs.

As reported before, no other studies have approached the combination effect of AT(N) profiles and NPSs under this point of view, when investigating conversion to dementia in MCI patients. Our study is a step forward in the way risk factors for conversion to dementia or poor prognosis are analyzed in MCI, as it helps to understand that the resulting effects of each factor do not necessarily have a monotonic effect, as it is assumed when no modulating effects are considered. This observation is especially relevant because the association between these two predictive factors (AT(N) profiles and NPSs) has been unclear to date. Meta-analytic data seem to not support a consistent association between AT(N) profiles in CSF and NPSs in MCI or AD dementia populations; however, depression is the only NPS to occasionally be linked with CSF levels [19]. Our results are consistent with this consideration of non-association, as depression, apathy, anxiety, apathy and irritability appeared to be homogeneously presented among AT(N) profiles. The study of Jang et al. (2020) is again concordant with this apparent disconnection between the AD biological profile and NPSs in MCI [14]. In this study, using data from the Alzheimer’s Disease Neuroimaging Initiative (ADNI), several NPS profiles exhibited different ratios of conversion to dementia but no association with the CSF AD-core biomarker status. However, a recent cross-sectional study [20] using a large sample of community-dwelling individuals reported that CSF Aβ42, T-tau/Aβ42 and p-tau/Aβ42 levels were associated with depression, and a prospective study including subjective cognitive decline (SCD) participants of the German Dementia Competence Network (DCN) study, CSF Aβ42 was found associated with an increment in depression severity over time [21].

Inconsistent results when determining the association between depression and AD-core biomarkers could be the consequence of several methodological reasons. One is the study design. Among the few studies focused on this topic, some are longitudinal and others cross-sectional. Populations from these studies are also heterogeneous, ranging from healthy individuals to patients with dementia. The operationalization of depression is also a potential limiting reason. Some studies are using clinical assessments, while others are based on the use of scales or historical data, for example. For more detail of the design of studies, see the compilation of Showraki [19]. The role and management of apathy–anhedonia is probably also contaminating the potential impact of depressive symptoms on the AD-core biomarkers, therefore not only impacting the frequently unraveled presentation of both conditions in clinical settings but also the way that both factors are analyzed (adjusted or not between them). Our goal was to add an observation emerged from results of the present study, as follows: depressive symptoms can be homogeneously distributed among the different AT(N) profiles, but the impact in prognosis (conversion to dementia) of this NPS may be substantially different along the different AT(N) profiles in MCI.

In this study, anxiety or irritability/lability have been also explored, but with a non-significant contribution in the prediction of conversion to dementia. The prevalence of anxiety among MCI patients has been reported in a very heterogeneous range of values [22], probably as a consequence of the recruitment strategies used and the methodology. Some studies have failed when finding an association in MCI with conversion to AD [23,24]. Others have found that this symptom has been presented as a risk factor for AD in population-based samples of MCI patients [25] but not in clinical samples [24]. Irritability has been described a relevant behavioral disturbance [26], with high rates of prevalence in MCI populations [27], and presented as a risk factor of conversion in MCI in the context of a memory unit [28]. Unfortunately, studies focused on irritability in MCI populations, exploring its impact on conversion ratios are scarce and any generalization is extremely difficult to assume. Further research with long-term follow-up in larger samples is needed to clarify the role of anxiety and irritability when predicting conversion to AD.

The *APOE* ε4 allele is by far the most robust genetic risk factor associated with sporadic AD [29]. In our study, though, the presence of the *APOE* ε4 status has a non-significant contribution in predicting the conversion to dementia in MCI patients when considering the AT(N) profile, which acts as a proxy of AD pathology. The predictive capacity of this genetic variable is only significant when comparing the *APOE* ε4 status between MCI converters and non-converters. A previous study of our group [28] showed that additive interactions combining NPSs (including depression) and *APOE* ε4 status emerged as consistent predictors of conversion to dementia in MCI. In the present study, the main effect of *APOE* ε4 status could not be considered relevant in the context of an adjusted model that included the AT(N) profiles. Additionally, in our study, no association was observed between the *APOE* ε4 status and any of the NPSs evaluated. The prevalence *APOE* ε4 carriers ranged from 57% in the prodromal AD AT(N) group to 11% in the normal AT(N) group (with intermediate values in the SNAP and brain amyloidosis AT(N) groups). Thus, we hypothesize that if an effect of the *APOE* ε4 status has to be sustained, this has to be driven through the impact of the AT(N) profiles on conversion to dementia. In MCI, the presence of the ε4 allele has been associated with lower levels of Aβ and elevated levels T-tau or p-tau in CSF compared to ε4 non-carriers [30]. This reported connection between *APOE* ε4 and AD-core biomarkers in CSF is not only concordant with our results, but a reinforcement of the previously proposed hypothesis. The modulating effects between these multifactorial factors should be explored in different and larger cohorts.

To date, few studies have explored the impact of the management of NPSs with specific treatments in early AD stages. This is a very important gap in scientific knowledge when investigating novel interventions that may modify the prognosis in MCI. A study showed decreased CSF Aβ42 levels in healthy elderly individuals after receiving a short treatment with antidepressant medication (escitalopram) [31]. Another study demonstrated that MCI patients with a history of depression presented a delay in progression to dementia after receiving treatment with selective serotonin reuptake inhibitors for several years [32]. More research is needed to identify potential therapeutic targets and their beneficial effects on preclinical and prodromal AD stages.

We acknowledge that our study has several limitations. One of the most important ones is the limited follow-up of the study. The identification of risk factors of conversion to dementia in MCI should be studied considering longer follow-ups than the two years analyzed in our study. Secondly, the spectrum of NPSs that are finally included in our analysis is limited. The main reason for excluding several of the NPSs available in the NPI-Q (elation/euphoria, eating/appetite problems, aberrant motor disturbances, delusions, hallucinations, agitation/aggression and disinhibition) was due to their low prevalence in our cohort. In MCI, however, these symptoms have low prevalence in general, and especially when considering data from a memory clinic [28]. A third limitation is related to the analysis design. Although survival analysis is a well-known strategy to identify risk factors of poor prognosis in preclinical and prodromal stages AD or other dementias, it usually consists of trying to find statistical connections between conditions assessed in a single time point (the baseline assessment) with an event observed beyond. The AT(N) profile is not necessarily a static condition, and neither are the patient’s neuropsychiatric status, comorbidities or other factors that are also considered in the analyses. Future longitudinal studies, based in more well-designed methodologies (for example, including several intermediate assessment points and using more sophisticated mathematical approaches) are required, not only to determine the relationship between AT(N) profiles and NPSs on prognosis, but also to elucidate the role and contribution of every factor in early stages of AD.

We have identified for the first time a synergic contribution of AT(N) profiles quantified in CSF and depressive symptoms on predicting conversion to dementia in MCI patients who were followed-up in a memory clinic. In particular, the combined effect of a CSF profile compatible with prodromal AD with depressive symptoms is associated with an exacerbation of the risk of conversion to dementia in MCI patients. These results are a step forward for the identification of risk factors of AD in the MCI stage, focusing simultaneously on both biochemical (CSF AD-core biomarkers) and behavioral (NPSs) conditions. Furthermore, this study helps to identify patients with MCI with worse prognosis and thus to accelerate their access to potential therapeutic interventions.

## 4. Materials and Methods

### 4.1. Study Participants

This study included 500 patients with a diagnosis of MCI who were evaluated at the memory clinic from of ACE Alzheimer Center Barcelona (single site) between 2016 and 2022. All participants underwent, within 5 months, a lumbar puncture (LP) for the quantification of CSF AD-core biomarkers and assessment of NPSs using NPI-Q [7].

### 4.2. Clinical Assessment

Study participants completed neurological, neuropsychological and social evaluations at the Ace Alzheimer Center Barcelona Memory Clinic. Patients were followed-up annually at the memory clinic. A consensus diagnosis was assigned to each patient by a multidisciplinary team of professionals [33]. Demographic information collected included age, sex and years of formal education. Cognitive assessment included the Spanish version of the Mini-Mental State Examination (MMSE) [34,35], the memory part of the Spanish version of the 7 Minute test [36], the Spanish version of the Neuropsychiatric Inventory Questionnaire (NPI-Q) [7], the Hachinski Ischemia Scale [37], the Blessed Dementia Scale [38] and the Clinical Dementia Rating (CDR) scale [39], as well as the comprehensive neuropsychological battery of ACE (N-BACE) [40]. MMSE [34,35] and NBACE [40] were assessed in all visits, while NPI-Q [7] was assessed at baseline.

### 4.3. NPSs Assessment

The NPI-Q [7] was used for the assessment of NPSs at baseline. This measure was administered by the neurologist/geriatrician during the clinical assessment, taking into account the information provided by the family member/caregiver and considering the patient’s situation in the last month. The NPI-Q includes the following NPSs: agitation/aggression, delusion, hallucination, depression/dysphoria, anxiety, euphoria/elation, apathy, disinhibition, irritability/lability, aberrant motor behavior, sleep, and eating/appetite. Every NPS was considered as present or absent, and only those with an occurrence >5% were included in the present study to provide consistent results (agitation/aggression (2.6%), delusion (2%), hallucination (1.4%), euphoria/elation (3%), disinhibition (3%), aberrant motor behavior (0.6%)). Thus, depression/dysphoria (50.4%), anxiety (43.6%), apathy (37.6%), irritability/lability (35.4%), and nighttime behaviors (20.6%) were the symptoms analyzed in the end.

### 4.4. MCI Subtypes

MCI patients were further classified as amnestic vs. non-amnestic and possible vs. probable subtypes [41]. An amnestic MCI subtype was assigned when memory deficits for the participant’s age and educational level, taking NBACE cut-offs [42], while a non-amnestic MCI subtype exhibited preserved memory but deficits in other cognitive domains [41]. The possible vs. probable MCI subtypes refer to the presence or absence, respectively, of comorbidities (such as cerebrovascular pathology, psychiatric and systemic disorders) which could explain or contribute to the cognitive deficits [43,44].

### 4.5. Conversion to Dementia

Participants who developed dementia were classified as MCI converters. The different underlying etiologies within the dementia group were classified according the following criteria: for AD, the 2011 NIA-AA for Alzheimer’s disease [45,46]; for vascular dementia (VaD), the National Institute of Neurological Disorder and Stroke and the Association Internationale pour la Recherche et l’Enseignement in Neurosciences criteria (NINDS-AIREN) [47] for frontotemporal dementia (FTD) [48] and for Lewy body dementia (LBD) [49]. Dementia conversion was defined using previously published criteria [8]. Participants who remained stable as MCI during the study follow-up period were classified as MCI non-converters.

### 4.6. Lumbar Puncture and Quantification of CSF Core Biomarkers for AD 

Lumbar punctures (LPs) were performed at Ace Alzheimer Center Barcelona by an experienced neurologist under fasting conditions. The collection protocol follows the recommendations of the Alzheimer’s Biomarkers Standardization [50]. CSF was collected passively in 10 mL polypropylene tubes (Sarstedt Ref 62.610.018) and centrifuged (2000× *g* 10 min at 4 °C) within 2 h of acquisition. After centrifugation, the fluid was aliquoted into polypropylene tubes (Sarstedt Ref 72.694.007) and stored at −80 °C until analysis. CSF biomarker results were not used for initial diagnostic endorsement in the memory clinic. The day of the analysis, one aliquot of 0.5 mL was thawed and used for the determination of Aβ1-42, total tau (T-tau) and p181-tau. Aβ and tau proteins were quantified by either the commercially available enzyme-linked immunosorbent assays (ELISAs) (INNOTEST, Fujirebio Europe, Göteborg, Sweden) (*n* = 252) or the chemiluminescense enzyme immunoassay (CLEIA) using the Lumipulse G 600 II automatic platform (Fujirebio Europe, Göteborg, Sweden) [51] (*n* = 248).

Using CSF biomarkers, participants were classified into four categories according to the AT(N) scheme [6]: normal AD biomarkers (A−T−N−), brain amyloidosis (A+T−N−), prodromal AD (including A+T+N−, A+T+N+ and A+T−N+) and suspected non-AD pathologic changes (SNAP, including A−T+N−, A−T−N+ and A−T+N+), where A refers to aggregated Aβ, T to aggregated tau and N to neurodegeneration or neuronal injury (Table 7).

Cut-offs from the Ace Alzheimer Center Barcelona CSF program were used to dichotomize each CSF biomarker into +/− as follows: for ELISA, Aβ1-42 < 676 pg/mL for A, p181-Tau > 58 pg/mL for T and T-Tau > 367 pg/mL for N; for CLEIA, Aβ1-42 < 796 pg/mL for A; p181-tau > 54 pg/mL for T and T-tau > 412 pg/mL for N. [8]. Clinicians were blinded to the CSF status of patients at the moment of the clinical assessment.

### 4.7. APOE Genotyping

Genomic DNA was extracted from peripheral blood using the commercially available Chemagic system (Perkin Elmer, Waltham, MA, USA). The *APOE* genotypes were extracted from the Axiom SP array (ThermoFisher, Waltham, MA, USA) [52,53]. A participant was defined as *APOE* ε4 carrier when at least one *APOE*ε4 allele was present.

### 4.8. Ethical Considerations

The LP consent was approved by the ethical committee of the Hospital Clinic i Provincial de Barcelona (Barcelona, Spain) in accordance with Spanish biomedical laws (Law 14/2007, 3 July, regarding biomedical research; Royal Decree 1716/2011, 18 November) and followed the recommendations of the Declaration of Helsinki.

### 4.9. Statistical Approach

Statistical analyses were performed using STATA 15 (Stata Corporation, College Station, TX, USA).

In order to explore the distribution of demographic and clinical variables between the four AT(N) groups, ANOVAs or Chi squared tests were executed. To determine the associations of demographic and clinical variables with the presence or absence of NPS, mean comparisons or Chi squared tests were used. This last strategy was also used to study the distribution of demographic and clinical variables between the MCI converters vs. MCI non-converters groups.

The main statistical approach focused on the analysis of the impact of the AT(N) profiles and NPSs on conversion to dementia and used Cox proportional hazard models. A first model included the mains effects of the AT(N) profiles (with the normal AT(N) group as the reference category) and the five NPSs. Age, sex, years of formal education, baseline MMSE, *APOE* ε4 carrier status and MCI subtype (amnestic–non amnestic) as well as MCI status (probable–possible) and MCI subtypes were also included in the model as adjusting factors.

To determine the effect of the combination of AT(N) profiles and NPSs on conversion to dementia, additive and multiplicative interactions were explored. A significant additive interaction means that the combined effect that is approached is larger (or smaller) than the sum of the two main exposures separately, while a multiplicative interaction assumes that the combined effect is larger (or smaller) than the product of the individual effects. Additive interactions have been assumed to be a better strategy to assess interaction effects, because they provide more applicable explanations for biological events than multiplicative interactions [54,55]. Due to the categorical AT(N) condition (four AT(N) groups) and binary (present/absent) NPSs, the comparison between normal and prodromal AD AT(N) groups were the focus of the analysis. The preparation of the data (creation of dummy variables) and the analytical codes for STATA reported by Van Der Weele and Knol for additive and multiplicative interactions and for categorical exposures were applied [56]. Cox proportional hazard models were used again, incorporating the new dummy variables, and adding the same adjusting factors previously reported. When a significant interaction was obtained, marginal means of cumulative hazard function for the combination of AT(N) profile and the corresponding NPSs were estimated and plotted for a better understanding of the accumulative effect of AT(N) and NPSs. Statistical testing was performed at a conventional two-tailed risk alpha at a level of *p* < 0.05.

## Figures and Tables

**Figure 1 ijms-24-01371-f001:**
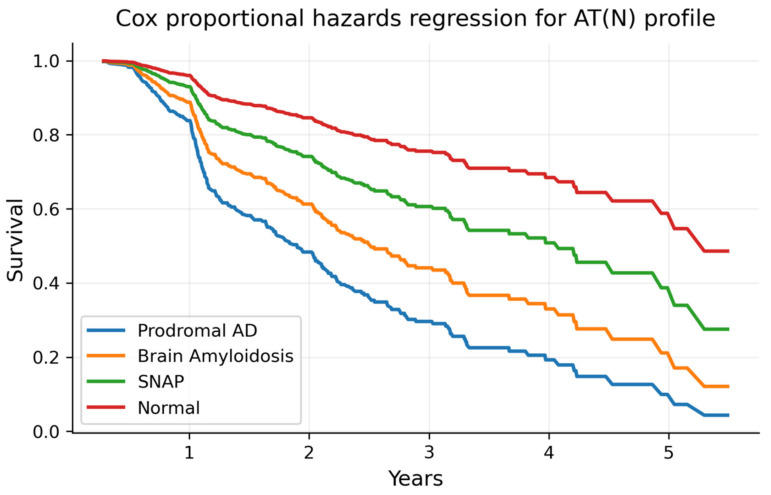
Survival curves of conversion to dementia for each AT(N) profile. Note: survival curves of conversion to dementia for each AT(N) profile obtained in the model presented in Table 5. AD: Alzheimer’s Disease; SNAP: Suspected Non-Alzheimer’s Pathology.

**Figure 2 ijms-24-01371-f002:**
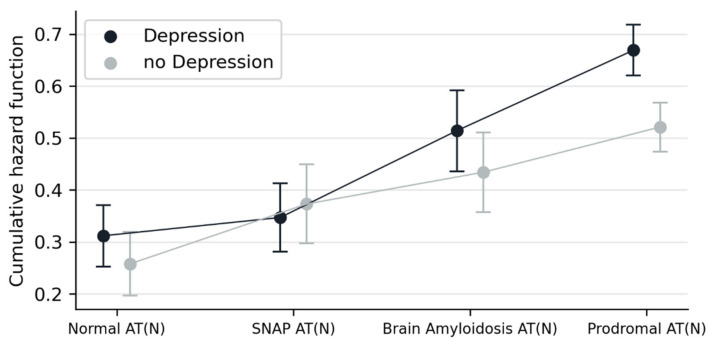
Estimated marginal means of cumulative hazard function when combining effects of AT(N) profiles and depression on conversion to dementia. Note: cumulative hazard function adjusted by age, sex, years of formal education, amnestic profile (yes/no), probable MCI (yes/no) *APOE* ε4 carrier (yes/no) and MMSE at baseline a MMSE: Mini-Mental State Examination; SNAP: Suspected Non-Alzheimer’s Pathology.

**Table 1 ijms-24-01371-t001:** Demographic and clinical characteristics of the study participants (*n* = 500).

	Mean (SD) or %
Age (y)	73.3 (7.5)
Female sex	55
Education (years)	7.84 (4.8)
Amnestic MCI profile	82.6
Probable MCI status	41.8
MMSE score	25.45 (3.3)
*APOE* ε4 carrier	36.4
AT(N) profiles	
Normal	26.8
SNAP	17.6
Brain amyloidosis	14.4
Prodromal AD	41.6
NPS	
Depression/Dysphoria	50.4
Anxiety	43.6
Apathy/Indifference	37.6
Irritability/Lability	35.4
Nighttime Behaviors	20.6
Time of follow-up (years)	2.08 (1.2)

AD: Alzheimer’s disease MCI: mild cognitive impairment; MMSE: Mini-Mental State Examination; NPSs: Neuropsychiatric Symptoms; SNAP: Suspected Non-Alzheimer’s Pathology.

**Table 2 ijms-24-01371-t002:** AT(N) profiles and their associations with demographic and clinical variables.

	Normal AT(N) (*n* = 134)	SNAP AT(N) (*n* = 88)	Brain Amyloidosis AT(N) (*n* = 72)	Prodromal AD AT(N) (*n* = 206)	*p* Value
Age (y)	70.18 (8.73)	73.94 (7.1)	72.94 (7.34)	75.18 (6.08)	<0.001
Female sex (%)	52.2	55.7	54.2	56.3	0.900
Education (y)	7.69 (3.76)	7.01 (4.19)	8.25 (7.53)	8.15 (4.43)	0.246
Amnestic MCI	73.1	84.1	79.2	89.3	0.001
Probable MCI	23.9	43.2	33.3	55.8	<0.001
MMSE score	26.09 (3.11)	25.42 (3.51)	25.56 (3.90)	25 (3.22)	0.026
*APOE* ε4 carrier	11.2	31.8	30.6	56.8	<0.001
NPS					
Depression	53	56.8	50	46.1	0.344
Anxiety	47	48.9	44.4	38.8	0.314
Apathy	40.3	31.8	40.3	37.4	0.595
Irritability/lability	37.3	33	33.3	35.9	0.896
Nighttime Behaviors	24.6	23.9	27.8	14.1	0.023

AD: Alzheimer’s disease; *APOE*: apolipoprotein E; MCI: mild cognitive impairment; MMSE: Mini-Mental State Examination; NPSs: Neuropsychiatric Symptoms; SNAP: Suspected Non-Alzheimer’s Pathology; y: years.

**Table 3 ijms-24-01371-t003:** NPS and their association with demographic and clinical variables.

	Depression (yes = 252/ no = 248)	Anxiety (yes = 218/ no = 282)	Apathy (yes = 188/ no = 312)	Irritability/Lability (yes = 177/no = 323)	Nighttime Behaviors (yes = 103/ no = 397)
Female sex	65.1/44.4 **	65.1/46.1 **	50/57.7	40.1/62.8 **	62.1/52.9
Age (y)	72.29 (7.6)/ 74.33 (7.23)	72.62 (7.77)/ 73.82 (7.22)	72.49 (7.19)/ 73.79 (79)	73.31 (7.26)/ 73.3 (7.61)	72.81 (7.59)/ 73.43 (7.46)
Education (y)	7.66 (5.13)/ 8.02 (4.42)	7.5 (5.37)/ 8.11 (4.29)	7.63 (4.22)/ 7.96 (5.11)	8.32 (4.32)/ 7.58 (5.02)	8.07 (6.55)/ 7.78 (4.23)
Amnestic MCI	79/86.1 *	80.7/84	88.3/79.2 *	85.3/81.1	83.5/82.4
Probable MCI	49.6/34.1 **	31.2/50 **	43.1/41	44.1/40.6	33/44.1
MMSE score	25.35 (3.18)/ 25.54 (3.31)	25.44 (3.08)/ 25.45 (3.37)	25.45 (3.19)/ 25.44 (3.28)	25.64 (2.96)/ 25.34 (3.339)	25.4 (3.28)/ 25.62 (3.14)
*APOE* ε4 carrier	39.1/33.7	36.7/36.1	36.7/36.2	41.2/33.7	38.8/35.8
AT(N) profiles					
Normal	28.2/25.4	28.9/25.2	28,7/25.6	28.2/26	32/25.4
SNAP	19.8/15.3	19.7/16	14.9/19.2	16.4/18.3	20.4/16.9
Brain Amyloidosis	14.3/14.5	14.7/14.2	15.4/13.8	13.6/14.9	19.4/13.1
Prodromal AD	37.7/44.8	36.7/44.7	41/41.3	41.8/40.9	28.2/44.6 *

* *p* < 0.05; ** *p* < 0.001. AD: Alzheimer’s disease; *APOE*: Apolipoprotein E; MCI: Mild Cognitive Impairment; MMSE: Mini-Mental State Examination; NPSs: Neuropsychiatric Symptoms; SNAP: Suspected Non-Alzheimer’s Pathology; y: years.

**Table 4 ijms-24-01371-t004:** Demographic and clinical overview of MCI non-converters vs. MCI converters.

	MCI Non-Converters (*n* = 276)	MCI Converters (*n* = 224)	*t* or Chi Statistics	*p* Value
Female sex (%)	53.3	56.7	0.589	0.443
Age (y)	71.72 (7.91)	75.24 (6.42)	5.49	<0.001
Education (y)	7.43 (5.52)	8.17 (4.09)	1.71	0.098
Amnestic MCI (%)	72.1	95.5	47.25	<0.001
Probable MCI (%)	31.2	54.9	28.67	<0.001
MMSE score	26.38 (2.81)	24.3 (3.38)	7.49	<0.001
*APOE* ε4 carrier (%)	29.7	44.6	11.91	0.001
Time follow-up (y)	2.51 (1.22)	1.51 (.93)	9.92	<0.001
AT(N) profiles (%)				
Normal	39.1	11.6		
SNAP	20.3	14.3		
Brain Amyloidosis	14.5	14.3		
Prodromal AD	26.1	59.8	71.64	<0.001
NPS (%)				
Depression	48.2	53.1	1.21	0.272
Anxiety	42.4	45.1	0.366	0.545
Apathy	33.3	42.9	4.78	0.029
Irritability/Lability	34.1	37.1	0.485	0.486
Nighttime Behaviors	21.7	19.2	0.489	0.484

AD: Alzheimer’s disease; *APOE*: Apolipoprotein E; MCI: Mild Cognitive Impairment; MMSE: Mini-Mental State Examination; NPSs: Neuropsychiatric Symptoms; SNAP: Suspected Non-Alzheimer’s Pathology; y: years.

**Table 5 ijms-24-01371-t005:** Cox regression analysis for AT(N) profiles, NPSs and adjusting factors.

	Hazard Ratio (HR)	z	95% CI HR	*p*-Value
AT(N) profiles				
SNAP	1.79	2.11	1.04–3.06	0.035
Brain Amyloidosis	2.92	3.96	1.72–4.97	<0.001
Prodromal AD	4.34	6.19	2.72–6.91	<0.001
NPS				
Depression	1.47	2.36	1.06–2.2	0.019
Anxiety	1.04	0.23	0.76–1.4	0.820
Apathy	1.4	2.32	1.05–1.86	0.020
Irritability/Lability	0.99	0.05	0.73–1.33	0.958
Nighttime Behaviors	0.86	0.84	0.59–1.22	0.403
Female sex	0.83	1.19	0.61–1.12	0.235
Age (y)	1.02	2.14	1.01–1.04	0.033
Education (y)	1.00	0.32	0.98–1.03	0.747
Amnestic MCI	3.63	3.88	1.89–6.67	<0.001
Probable MCI	1.51	2.75	1.12–2.02	0.006
MMSE score	0.87	6.22	0.83–0.91	<0.001
*APOE* ε4 carrier	0.98	0.12	0.73–1.31	0.904

For AT(N) profiles, the Normal group is the reference category. AD: Alzheimer’s Disease; *APOE*: Apolipoprotein E; CI: Confidence Interval; MCI: Mild Cognitive Impairment; MMSE: Mini-Mental State Examination; NPSs: Neuropsychiatric Symptoms; SNAP: Suspected Non-Alzheimer’s Pathology; y: years.

**Table 6 ijms-24-01371-t006:** Additive and multiplicative interactions in Cox regression analysis for ATN and NPSs.

	Additive Interactions			Multiplicative Interactions		
	Coeff.	95% CI Coeff.	*p*	Coeff.	95% CI Coeff.	*p*
AT(N)*Depression	3.23	0.20–6.26	0.037	0.023	−0.84–0.88	0.958
AT(N)*Anxiety	1.63	−0.01–3.28	0.051	1.73	0.72/4.13	0.213
AT(N)*Apathy	1.33	−0.99/3.66	0.260	0.86	0.36/2.02	0.735
AT(N)*Irritability/Lability	0.81	−1.30/1.87	0.728	1.27	0.52/3.11	0.591
AT(N)*Nighttime Behaviors	−2.24	−5.43/0.94	0.167	0.36	0.13/0.94	0.034

Additive and multiplicative interactions are calculated contrasting normal AT(N) and prodromal AD AT(N) groups. Results are adjusted by age, sex, years of education, amnestic profile (yes/no), probable MCI (yes/no) *APOE* ε4 carrier (yes/no) and MMSE at baseline.

**Table 7 ijms-24-01371-t007:** Biomarkers results included in each ATN category.

ATN Categories	Aβ-42	p181-tau	Total Tau
Normal	−	−	−
SNAP	−	+	+
SNAP	−	+	−
SNAP	−	−	+
Brain Amyloidosis	+	−	−
Prodromal AD	+	+	−
Prodromal AD	+	−	+
Prodromal AD	+	+	+

## Data Availability

The datasets generated and/or analyzed for this study will be made available by the corresponding author upon reasonable request.

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
