# Peer review of "The Synergic Effect of AT(N) Profiles and Depression on the Risk of Conversion to Dementia in Patients with Mild Cognitive Impairment"

_ijms, 2023, doi:10.3390/ijms24021371_

Round 1

Reviewer 1 Report

Marquie et al. presented the synergic effect of AT(N) profiles and depression on the risk of conversion to dementia in patients with MCI. This is an interesting paper that identifies risk factors of AD in the MCI stage by focusing simultaneously on both biochemical and behavioral conditions. The paper is well written. There are, however, some issues to be addressed to further improve the manuscript.

1.     As the authors themselves pointed out as a limitation, the two-year follow-up is too short to discuss the transition from MCI to dementia. What is the rationale for the two-year follow-up?

2.     The period of compulsory education in Spain is supposed to be 10 years. In the present study, average education period was 7.84 years. Could the level of education in patients have an impact on the results?

3.     The cognitive-assessment of patients were followed-up annually by using MMSE and NBACE in the present study. Does it enough to evaluate the conversion to dementia?

Is imaging analysis with CT or MRI not necessary for more precise diagnosis?

Author Response

review1

Comments and Suggestions for Authors

Marquie et al. presented the synergic effect of AT(N) profiles and depression on the risk of conversion to dementia in patients with MCI. This is an interesting paper that identifies risk factors of AD in the MCI stage by focusing simultaneously on both biochemical and behavioral conditions. The paper is well written. There are, however, some issues to be addressed to further improve the manuscript.

  1. As the authors themselves pointed out as a limitation, the two-year follow-up is too short to discuss the transition from MCI to dementia. What is the rationale for the two-year follow-up?

**The ACE Alzheimer Center Barcelona (ACE) CSF program started in 2016 as an essential research activity. Obviously, those patients included at the starting moment of the program are currently exposed to longer period of follow-up, but the incidence of new MCI cases enriching this total sample is increasing year by year, presenting shorter follow-ups. This is the result of the intrinsic enrollment strategy of our Memory Unit. As a reference, in a sample of close to 2200 MCI patients, using only clinical variables as predictors of conversion, but under a comparable study design, it was reported a mean of follow-up of 2.2 years (Roberto et al., 2021).

  1. The period of compulsory education in Spain is supposed to be 10 years. In the present study, average education period was 7.84 years. Could the level of education in patients have an impact on the results?

**The compulsory education of 10 years is for the last Spanish generations, but our MCI (mean age of 73) were mostly schooled in the 50-60s. In Spain, as an example, and due to the great recession after the civil war, illiteracy rates remains exceptionally high for an European country among people older than 65 years, in special for women (Rosende-Roca et., 2021). According to results of the present study, years of formal education has no statistical impact on conversion, adjusted or not by the rest of factors, as reported in tables 5 & 6 of the manuscript.

  1. The cognitive-assessment of patients were followed-up annually by using MMSE and NBACE in the present study. Does it enough to evaluate the conversion to dementia?

**MMSE & NBACE are assessed in every contact with the patient, but the diagnose procedure dependents of the information collected by the clinical assessment of neurologists, the neuropsychological explorations of neuropsychologist (using the NBACE), and the functional and social condition according to data provided by our social workers. Every exploration, annually programmed, applies the same protocol. This multifactorial decision protocol is discussed by all the professionals involved in the same case, determining the current diagnose of a patient in a consensus way. In fact, MMSE has an insignificant relevance in this protocol. We have rewritten the method section to improve this description.

Is imaging analysis with CT or MRI not necessary for more precise diagnosis?

**MRI, when available, can be use in the decision-making process. Clinicians are blind to the CSF status of patients when determining diagnose. We have added this information in the method section.

review2

In this study, 500 patients were used to analysis determine risk of conversion, they also considered the additive and multiplicative interactions between AT(N) profile and NPS on the conversion to dementia. They collected a lot of data from their analysis. However, authors should prepare a major revision to address the below comments for publish in second review.

  1. In results, authors showed 55% of patients were female, but female didn’t show significant in Table 3 and Table 5. How about the male in their study?

**Tables 3&5 report only the label -female- and their %, but men are also included in all analysis. Women and men are homogeneously distributed among ATN conditions (p=443; Table 3) and women have comparable risk effect than men when predicting conversion to dementia (p=.235; Table 5).

  1. In analysis of AT(N) profiles and their associations with demographic and clinical variable, authors showed the depression, anxiety, and apathy were not significant (P>0.05, Table 3). should authors discuss these factors in At(N) profiles associations?

**Yes. We have discussed now these non-significant results (lines 229-231).

  1. Like the above mentioned, should authors discuss the no-significant factors in their study? (Table5, 6)

**Sure. We have improved the discussion in this sense (lines 255-267)

Reviewer 2 Report

In this study, 500 patients were used to analysis determine risk of conversion, they also considered the additive and multiplicative interactions between AT(N) profile and NPS on the conversion to dementia. They collected a lot of data from their analysis. However, authors should prepare a major revision to address the below comments for publish in second review.

1.   In results, authors showed 55% of patients were female, but female didn’t show significant in Table 3 and Table 5. How about the male in their study?

2.   In analysis of AT(N) profiles and their associations with demographic and clinical variable, authors showed the depression, anxiety, and apathy were not significant (P>0.05, Table 3). should authors discuss these factors in At(N) profiles associations?

3.   Like the above mentioned, should authors discuss the no-significant factors in their study? (Table5, 6)

Author Response

review2

In this study, 500 patients were used to analysis determine risk of conversion, they also considered the additive and multiplicative interactions between AT(N) profile and NPS on the conversion to dementia. They collected a lot of data from their analysis. However, authors should prepare a major revision to address the below comments for publish in second review.

  1. In results, authors showed 55% of patients were female, but female didn’t show significant in Table 3 and Table 5. How about the male in their study?

**Tables 3&5 report only the label -female- and their %, but men are also included in all analysis. Women and men are homogeneously distributed among ATN conditions (p=443; Table 3) and women have comparable risk effect than men when predicting conversion to dementia (p=.235; Table 5).

  1. In analysis of AT(N) profiles and their associations with demographic and clinical variable, authors showed the depression, anxiety, and apathy were not significant (P>0.05, Table 3). should authors discuss these factors in At(N) profiles associations?

**Yes. We have discussed now these non-significant results (lines 229-231).

  1. Like the above mentioned, should authors discuss the no-significant factors in their study? (Table5, 6)

**Sure. We have improved the discussion in this sense (lines 255-267)